# Reduction in CgA-Derived CST Protein Level in HTR-8/SVneo and BeWo Trophoblastic Cell Lines Caused by the Preeclamptic Environment

**DOI:** 10.3390/ijms24087124

**Published:** 2023-04-12

**Authors:** Michalina Bralewska, Tadeusz Pietrucha, Agata Sakowicz

**Affiliations:** Department of Medical Biotechnology, Medical University of Lodz, Zeligowskiego 7/9, 90-752 Lodz, Poland

**Keywords:** chromogranin A, catestatin, pregnancy, preeclampsia, hypertension

## Abstract

One of the most dangerous complications of pregnancy is preeclampsia (PE), a disease associated with a high risk of maternal and fetal mortality and morbidity. Although its etiology remains unknown, the placenta is believed to be at the center of ongoing changes. One of the hormones produced by the placenta is chromogranin A (CgA). Thus far, its role in pregnancy and pregnancy-related disorders is enigmatic, yet it is known that both CgA and its derived peptide catestatin (CST) are involved in the majority of the processes that are disturbed in PE, such as blood pressure regulation or apoptosis. Therefore, in this study, the influence of the preeclamptic environment on the production of CgA using two cell lines, HTR-8/SVneo and BeWo, was investigated. Furthermore, the capacity of trophoblastic cells to secrete CST to the environment was tested, as well as the correlation between CST and apoptosis. This study provided the first evidence that CgA and CST proteins are produced by trophoblastic cell lines and that the PE environment has an impact on CST protein production. Furthermore, a strong negative correlation between CST protein level and apoptosis induction was found. Hence, both CgA and its derived peptide CST may play roles in the complex process of PE pathogenesis.

## 1. Introduction

Pregnancy is a unique state in which the body must undergo multiple changes to prepare itself for the developing fetus. Any failure in the adaptation process can have serious ramifications for the health of both the mother and the baby.

One of the most dangerous complications of pregnancy is preeclampsia (PE), a disorder associated with a high risk of maternal and fetal mortality and morbidity. The diagnostic criteria for PE comprise an increase in blood pressure to ≥140 mmHg systolic or ≥90 mmHg diastolic in previously normotensive women after at least 20 weeks of gestation. Moreover, at least one of the following symptoms must occur: proteinuria, serum creatinine level >1 mg/dL, hemolysis, thrombocytopenia, elevated transaminase levels, neurological disorders or uteroplacental dysfunction (e.g., fetal growth restriction) [1].

Although the etiology of this phenomenon remains unknown, both maternal and placental factors are believed to play a significant role in its development. One of the most widely accepted theories of PE proposes that shallow trophoblast invasion into maternal decidua forces placental cells to grow in hypoxic conditions (1–2% of oxygen), leading to uncontrolled placental apoptosis and fetal malnutrition [2]. This results in the appearance of multiple placental factors, such as soluble fms-like tyrosine kinase 1 (sFlt-1) or soluble endoglin (sEng), in the maternal circulation [3]. These factors act on the maternal vascular endothelium, inducing oxidative stress and the production of reactive oxygen species (ROS), e.g., superoxide (O_2_^•−^) and hydrogen peroxide (H_2_O_2_); they also stimulate the production and secretion of pro-inflammatory cytokines (e.g., tumor necrosis factor α; TNFα, interleukin 6; IL6) into the maternal blood [2,4,5].

However, since the placenta plays a major role in the pathogenesis of PE, the etiology may also have an endocrine basis. Being an endocrine organ, the placenta produces and releases hormones and active molecules with a direct impact on the course of pregnancy (e.g., progesterone, oestrogens) [6]. Although most of those hormones have been well studied, several molecules have an uncertain function. One such hormone is chromogranin A (CgA). Although it is generally secreted by the chromaffin cells of the adrenal medulla, it has been identified immunohistochemically in the syncytiotrophoblastic layer of the placenta [7,8]. CgA is considered, inter alia, an anti-apoptotic factor; an in vitro study on prostate cancer (PC) cells identified increased survival of PC cells through Akt-mediated survivin up-regulation [9]. Furthermore, CgA is believed to influence blood pressure (BP) regulation with one of its derived peptides, catestatin (CST), and is co-stored and co-released with the catecholamines in chromaffin granules, which have also been implicated in BP dysregulation.

CgA concentrations are also used as diagnostic markers in patients with hypertensive pheochromacytoma [10]. Additionally, in both neoplastic tumors and PE, the only known way to lower the level of catecholamines co-secreted with CgA is to remove the main source, i.e., the tumor in pheochromacytoma patients and placenta in preeclamptic women [3,11].

Although the presence of CgA in the placenta has been confirmed, little is known of its impact on pregnancy or pregnancy-related disorders. Additionally, no data exist regarding the role of the CgA-derived peptide CST in pregnancy or PE, nor of its presence in the cell cultures representing human trophoblast. Although various studies on, inter alia, apoptosis, hypoxia and oxidative stress have been performed on trophoblastic cell lines, none have examined whether those cells are able to produce CgA and secrete CST [4,12,13].

Therefore, this study investigates the influence of the preeclamptic environment on the production of CgA using two cell lines: HTR-8/SVneo and BeWo. The two cell lines reflect different stages of maturity in the human placenta: HTR-8/SVneo represents a human trophoblast from the first trimester of gestation, and BeWo from the third trimester [14,15]. The study also determines the capacity of trophoblastic cells to secrete CST to the environment, thus establishing whether placental cells are able to help maintain correct BP over the course of gestation by producing and secreting CST to the maternal bloodstream. As CgA is also linked with the process of apoptosis, the study will also examine the correlation between CST level and apoptotic index of placental cells.

## 2. Results

Preeclamptic conditions were replicated using two stimulants, viz., IL6 and H_2_O_2_, reflecting the respective inflammation and oxidative stress present in the PE placenta. The stimulants were chosen based on previous studies indicating that both are known to be elevated in PE patients [16,17]. Hypoxia was generated in a hypoxia chamber, ensuring constant, long-lasting culture in 2% of oxygen. Precise selection of the oxygen conditions is of key importance to the whole study. As noted by Pavlacky and Polak, an improper oxygen level can result in various multiple cellular responses influencing cell signaling, migration, proliferation and apoptosis [18]. Therefore, instead of the 21% oxygen concentration typically used for this experiment, 8%O_2_ was used for the normoxic controls, as this value corresponds to the physiological trophoblast oxygen level [19]. Additionally, instead of short-term stimulation, the cultures were maintained for 96 h to best imitate the process ongoing in the PE placenta. Each 24 h of culture in stimulating medium corresponds to ten weeks of gestation; therefore, the 96 h used in this study corresponds to the natural 40-week duration of gestation.

### 2.1. Preeclamptic Conditions Have a Minimal Influence on CHGA Gene Expression and CgA Protein Level in HTR-8/SVneo and BeWO Cells

In HTR-8/SVneo cells kept in a preeclamptic environment created by hypoxia and supplementation of the cell culture medium with IL6 and H_2_O_2_, the expression of the gene coding for chromogranin A was lowered. Remained stimulation variants (i.e., HC—Hypoxic conditions, ILPE1—Inflammatory-like preeclamptic environment 1, ILPE2—Inflammatory-like preeclamptic environment 2, OLPE1—Oxidative stress-like preeclamptic environment 1 and OLPE2—Oxidative stress-like preeclamptic environment 2) did not influence CHGA gene expression compared to controls (Figure 1).

The HTR-8/SVneo cells cultured in hypoxia (2%O_2_), and treated with combinations of IL6 and H_2_O_2_, demonstrated reduced *CHGA* gene expression (*p* < 0.001) compared to controls, i.e., cells incubated in normoxia (8%O_2_) in a medium with 2.5%FBS without stimulants.

**Figure 1 ijms-24-07124-f001:**
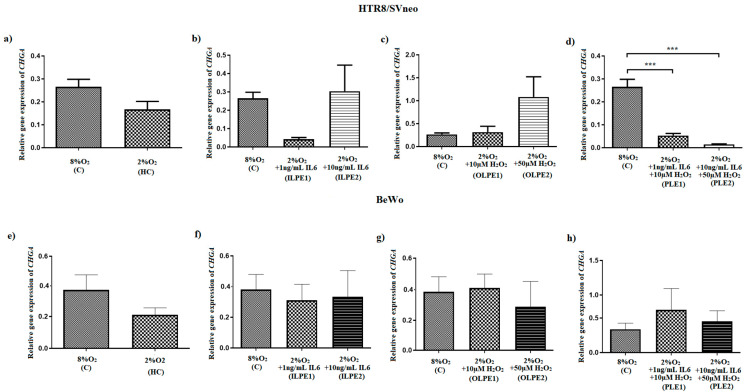
A comparison of relative CHGA gene expression between HTR-8/SVneo (**a**–**d**) and BeWo cells (**e**–**h**) under different stimulation regimens. The results are expressed as mean ± SEM (*n* = 3). The results from the cells cultured in 8%O_2_ (control) and 2%O_2_ were compared using Student’s *t*-test. One-way ANOVA with post hoc Bonferroni’s correction was used to compare the variants and the controls. *p* < 0.05 was considered as significant, *** *p* < 0.001 as compared to control. C—Control (8%O_2_), HC—Hypoxic conditions (2%O_2_), ILPE1—Inflammatory-like preeclamptic environment 1 (2%O_2_ + 1 ng/mL IL6), ILPE2—Inflammatory-like preeclamptic environment 2 (2%O_2_ + 10 ng/mL IL6), OLPE1—Oxidative stress-like preeclamptic environment 1 (2%O_2_ + 10 µM H_2_O_2_), OLPE2—Oxidative stress-like preeclamptic environment 2 (2%O_2_ + 50 µM H_2_O_2_), PLE—Preeclamptic-like environment 1 (2%O_2_ + 1 ng/mL IL6 + 10 µM H_2_O_2_), PLE2—Preeclamptic-like environment 2 (2%O_2_ + 10 ng/mL IL6 + 50 µM H_2_O_2_). It was found that the production of CgA protein by the trophoblastic cells depended on the environment. All stimulated variants of both HTR-8/SVneo and BeWo cells demonstrated lowered CgA levels compared to controls. However, only the HTR-8/SVneo cell line yielded significantly different outcomes (*p* < 0.01). Stimulation with variants reflecting a more or less-intense preeclamptic-like environment resulted in lower relative CgA protein levels compared to the control gene, i.e., GAPDH, and compared to the control culture (8%O_2_; normoxia) (Figure 2B). Additionally, CgA protein level was found to be depleted by hypoxia (2% of oxygen), independently of the applied stimulants (Figure 2A). In addition, no significant difference in relative CgA protein level was observed between the other stimulants and controls, in either HTR-8/SVneo or BeWo cells.

**Figure 2 ijms-24-07124-f002:**
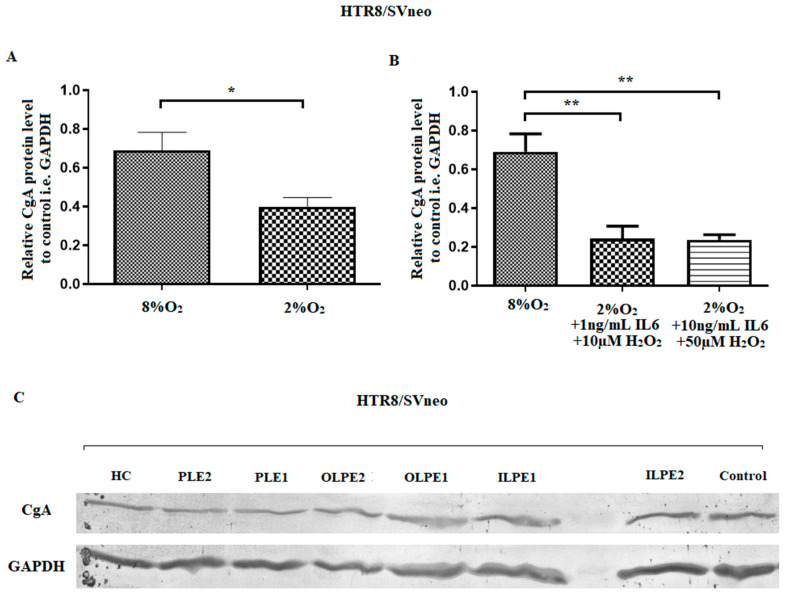
Relative CgA protein level compared to GAPDH level (housekeeping gene) in HTR-8/SVneo cells under different stimulation regimens, as indicated by Western blot analysis. (**A**) All cells were cultured in medium with 2.5%FBS in normoxic (control) or hypoxic conditions. The 8%O_2_ (control) and 2%O_2_ cultures were compared using the Student’s *t*-test. (**B**) CgA expression in cells incubated in medium including IL6 and H_2_O_2_ under hypoxia (preeclamptic) conditions were compared with untreated controls using one-way ANOVA with Bonferroni’s correction. The results are expressed as mean ± SEM (*n* = 3). The level of significance was assumed as *p* < 0.05; ** *p* < 0.01, * *p* < 0.05 compared to control. (**C**) The representative results of Western blot analysis showing bands for CgA and the control GAPDH protein; C—Control (8%O_2_), HC—Hypoxic conditions (2%O_2_), ILPE1—Inflammatory-like preeclamptic environment 1 (2%O_2_ + 1 ng/mL IL6), ILPE2—Inflammatory-like preeclamptic environment 2 (2%O_2_ + 10 ng/mL IL6), OLPE1—Oxidative stress-like preeclamptic environment 1 (2%O_2_ + 10 µM H_2_O_2_), OLPE2—Oxidative stress-like preeclamptic environment 2 (2%O_2_ + 50 µM H_2_O_2_), PLE—Preeclamptic-like environment 1 (2%O_2_ + 1 ng/mL IL6 + 10 µM H_2_O_2_), PLE2—Preeclamptic-like environment 2 (2%O_2_ + 10 ng/mL IL6 + 50 µM H_2_O_2_).

### 2.2. Hypoxia, Inflammation and Oxygen Stress Lower CST Protein Level in HTR-8/SVneo and BeWO Cells

Hypoxic conditions (2% of oxygen) significantly lowered CST protein level (ng/mL of medium) in HTR-8/SVneo cells compared to control (8% of oxygen). Additionally, both stimulants, viz., IL6 representing an inflammatory environment and H_2_O_2_ an oxidative environment, resulted in a significant depletion of CST protein level in hypoxic conditions compared to controls (Figure 3A). Stimulation with varying concentrations of preeclamptic factors (IL6 and H_2_O_2_) under hypoxic conditions resulted in lower CST protein levels in cultured HTR-8/SVneo cells as compared to controls (normoxia).

For the BeWo cell line kept in hypoxic conditions, treatment with Il-6 and H_2_O_2_, and with PLE, caused a depletion in CST protein level compared to the unstimulated cells cultured in normoxia (8% of oxygen) (Figure 3B).

### 2.3. Preeclamptic Conditions Induce Apoptosis—The Higher Apoptotic Index Correlates with a Low Level of Catestatin Both for HTR-8/SVneo and BeWo Cell Lines

The results of The Muse™ Annexin V with Dead Cell Assay Kit for each stimulation variant, both for HTR-8/SVneo and BeWo cell line, are presented in Figure 4. Further, statistical analysis indicated that the HTR-8/SVneo cells, treated with 50 µM of H_2_O_2_ and both doses of mixed stimulants caused a significant increase in the number of apoptotic cells; however, the most remarkable increase compared to controls (*p* < 0.001) was obtained in hypoxic cells treated with higher doses of both stimulants. The reminded stimulants do not influence apoptotic index in HTR-8/SVneo cells. For the BeWo cell line, statistically significant results were obtained for each stimulation variant. The most significant apoptosis induction (*p* < 0.001) was observed in cells treated with 10 µg/mL of IL6, 10 µM and 50 µM of H_2_O_2_ and with both mixes of stimulants, in hypoxic conditions, as compared to the control. Additionally, a low level of oxygen (2%) alone was sufficient to cause an increase in the number of apoptotic cells (*p* < 0.01) in BeWo, as compared to the unstimulated control.

Furthermore, we observed that the Spearman’s rank order correlation exists between the catestatin level and apoptotic index for both HTR-8/SVneo and BeWo cells (Figure 5).

## 3. Discussion

The aim of the present study was to examine whether a preeclamptic environment, i.e., one rich in inflammatory and oxidative stress factors, has an impact on *CHGA* gene expression and CgA protein level. It also investigated the capacity of trophoblastic cell lines to secrete CST. Our findings also suggest that CgA and CST may play parts in the complex process of PE pathogenesis. The main advantage of the presented work is that the whole experiment was carried out under long-lasting hypoxic conditions (2%O_2_), and its findings are compared with those from trophoblast under physiological conditions, corresponding to 8%O_2_ [19]. This time period more accurately reflects the long period from the beginning of the pathological process in the placenta until the appearance of clinical symptoms of PE; indeed, placental cells typically live in hypoxic conditions characterized by high levels of inflammatory factors and oxidative stress. Our findings not only provide the first evidence that CgA and CST are present in human trophoblastic cells lines, but also that preeclamptic conditions result in the depletion of the CST protein level in the cells. Until now, only a few studies have investigated the role of CgA in pregnancy, and even fewer have examined the role of its derived peptide CST. In addition, these studies have focused more on clinical observations and fail to offer sufficient proof that trophoblastic cells are capable of secreting CST into their environment.

Although many of the placental and maternal factors engaged in PE pathogenesis are well studied, some of them remain unclear. One of such molecules is CgA, found to be produced by the placenta.

CgA is a 49 kDa acidic polypeptide belonging to the granin family. The main functions of CgA are the biogenesis, storage, trafficking and release of catecholamines from the dense core secretory granules of adrenergic neurons [20]. Depending on the site of cleavage, CgA derived peptides can act both as vasoconstrictors or vasodilators, regulating blood pressure [21,22]. One such vasodilator is CST, a peptide which is also involved in the regulation of oxidative stress and chronic inflammation [23,24,25].

At the beginning of pregnancy, placental cells live under hypoxic conditions (1–2%O_2_) [19]. This promotes proliferation and enables the differentiation of trophoblastic cells into invasive cell subtypes. At the same time, a local inflammatory state develops, supporting matrix degradation and proper implantation. As the implantation proceeds, the oxygen level in the environment increases and the inflammatory reactions gradually slow down [19,26]. In PE, this process is disrupted. Failures in vessel remodeling lead to a prolonged state of hypoxia.

Hence, the present article examines the impact of hypoxia, inflammation and oxidative stress on *CHGA* gene expression, CgA and CST protein levels, and apoptosis in two trophoblastic cell lines characteristic of the first (HTR-8/SVneo) and third trimesters of gestation (BeWo). Our previous studies, based on the human placenta, showed an increase in *CHGA* gene expression in PE patients, accompanied by non-significant differences in CgA protein level between study and control groups. Therefore, the next step was to confirm whether trophoblast cell lines are characterized by the same relationship [27].

Apart from the two stimulation variants reflecting a preeclampsia-like environment (PLE1: 2%O_2_ + 1 ng/mL IL6 + 10 µM H_2_O_2_ and PLE2: 2%O_2_ + 10 ng/mL IL6 + 50 µM H_2_O_2_), no significant changes in *CHGA* gene expression were obtained in the HTR-8/SVneo and BeWO cell lines compared to controls (Figure 1). Despite this, however, both variants demonstrated reduced CgA protein levels compared to controls (Figure 2B). In addition, hypoxic conditions alone resulted in a depletion of the CgA protein.

Although the two sets of outcomes (from the cell cultures and placental samples) are not contradictory, differences were noted between the given outcomes and those obtained from placental tissue; this may be a consequence of the obvious disparity in the types of tested material, the main one being that cell cultures will never entirely reflect processes ongoing in human organism. Furthermore, all the statistically significant outcomes were obtained for the HTR-8/SVneo cell line, i.e., representing the first trimester human trophoblast, whereas placental samples are usually collected at the end of gestation, i.e., the third or late second trimester.

Sahu et al. report that targeted ablation of *CHGA* gene in knockout mice *chga^−/−^*, which is directly connected with CST insufficiency, can cause severe arterial hypertension [28]. Therefore, the presence of high CgA levels may be a part of maternal compensatory mechanisms. As mentioned before, at the beginning of pregnancy (here represented by the HTR-8/SVneo cells), low oxygen level and mild inflammation are necessary for proper implantation. Therefore, the fact that significantly lower CgA protein level was observed between cells incubated in 8%O_2_ and 2%O_2_ suggests that trophoblastic cells both from early normotensive and preeclamptic pregnancy are characterized by a low production of CgA. This low CgA production by early trophoblast is additionally depleted in hypoxia complicated by inflammatory conditions (ILPE1, i.e., 2%O_2_ + 1 ng/mL IL-6 + 10 µM H_2_O_2_ and ILPE2, i.e., 2%O_2_ + 10 ng/mL IL-6 + 50 µM H_2_) in comparison to 8%O2. However, this level is insignificant lower between cells incubated only in 2%O_2_ vs. ILPE1 or 2%O_2_ vs. ILPE2 (results not shown). Furthermore, in the present study, CHGA expression and CgA protein level did not significantly differ between the variants representing the third trimester trophoblast, i.e., the BeWo cell line. Interestingly, our previous study conducted on placental samples obtained at the term of gestation found only CHGA expression level, but not protein, elevated in PE patients compared to healthy controls [27]. This may suggest that together, the course of preeclamptic pregnancy, maternal organism attempts to increase the level of CgA protein by overexpression of its gene; however, this mechanism might be ineffective or the full protein CgA is immediately digested into smaller peptides that can lead to the adverse pregnancy outcomes.

CST is a small peptide derived from CgA. It is characterized, among others, by its pro-angiogenic and anti-inflammatory effects, hypotensive way of action, and its role in the defense against oxidative stress-induced apoptosis [29,30,31]. Although the exact mechanism by which CST reduces infiltration by monocytes and macrophages remains unknown, it may be connected with its regulation of chemotaxis [32]. CST has also been found to demonstrate pro-angiogenic potential in vitro and in vivo. Theurl et al. report that CST acts on endothelial cells, releasing fibroblast growth factor and stimulating its signaling, and that CST induced angiogenesis in the cornea neovascularization assay in mice, resulting in, inter alia, increased blood perfusion [31]. CST level is also known to be lower in hypertensive patients, as confirmed in vivo on CST knockout mice (CST-KO) developing hypertension [29,33,34]. The anti-hypertensive effects of CST were shown to be mediated, at least in part, by its anti-inflammatory potential, and by its induction of a massive release of histamine and subsequent nitric oxide (NO) production, leading to vasodilatation [35].

Hypertension, chronic inflammation and hypoxia are key characteristics of PE. In the present study, an inflammatory-like preeclamptic environment (ILPE), imitated by in vitro IL6 stimulation, caused a depletion in CST protein level produced by the trophoblastic cells. Similar results were obtained following stimulation with H_2_O_2_, reflecting an oxidative stress-like preeclamptic environment (OLPE); in both environments, the cells were cultured in hypoxic conditions. Considering the pro-angiogenic, anti-inflammatory and anti-hypertensive role of CST [36], the obtained outcomes accurately reflect the main features observed for PE. They also stay in line with the majority of literature, where CST protein level is found to be diminished in hypertensive patients [33,34,36]. In contrast, Tüten et al. report that CST protein level was increased in the serum of preeclamptic women [37]. This contrasts with our previous studies on PE tissue, in which CST protein levels were found to be diminished in PE patients compared to the healthy controls [27], as in the present study; however, Tüten et al. emphasize that their results are not consistent with the rest of the literature. This inconsistency may be due to the difference in CST protein production observed during different stages of pregnancy; alternatively, its production may be influenced by maternal compensatory mechanisms, or the CST level in the serum of PE mothers may be derived from other origins than the placenta, resulting in a different systemic protein level to that present in the placenta. Furthermore, Tüten et al. include additional disease entities, e.g., renal insufficiency, in the inclusion criteria, where renal disease itself is connected with increased plasma CST level [38].

Significant differences in the intensity of apoptotic processes occur in the placenta depending on the stage of pregnancy. Throughout gestation, oxygen levels and apoptosis must be modulated to support proper placental invasion, cytotrophoblast fusion and syncytiotrophoblast function. However, in PE, prolonged hypoxia, ROS and excessive inflammation can lead to uncontrolled and excessive cell death [39].

In the present study, the complete preeclamptic environment, as well as the presence of hypoxia, ILPE or OLPE alone, caused induction of apoptosis in the tested cell lines. More statistically significant outcomes were obtained in the BeWo cells. This is probably connected with the characteristics of the two cell lines. HTR-8/SVneo cells are representative of the first trimester trophoblast, characterized by a physiological low oxygen level and mild inflammation. Therefore, apoptosis was induced only by the highest doses of stimulants, reflecting OLPE2, and both combinations reflecting a preeclamptic-like environment, viz., PLE1 and 2. In contrast, the BeWo cells are reflective of the third trimester, and normally grow in 8% of oxygen; therefore, the presence of a hypoxic environment, and each stimulation variant, resulted in significant apoptosis induction. The above scheme accurately reflects the processes ongoing in the preeclamptic placenta of pregnant women.

Furthermore, in the presented study, CST protein level was negatively correlated with apoptosis induction in both cell lines (Figure 5). This strong connection between CST and apoptosis suggests it may have a possible influence on one of the main features of PE. Liao Feng et al. report that CST protects cells from death, thus attenuating oxidative stress-mediated apoptosis [40]. This action is mediated by the ERK1/2 and PI3 K/Akt pathways. This is in accordance with Song-Yun Chu et al., in which CST was shown to regulate the PI3K/Akt pathway by activation of the β2 adrenergic receptor (β2) [30]. Both the PI3 K/Akt and ERK1/2signaling pathways are engaged in various processes of pregnancy, regulating, among other things, decidualization, implantation and embryogenesis, as well as various cellular processes including cell growth, proliferation, migration and survival [41,42,43].

Activation of the β2 receptor results in increased vasodilation via endothelium-dependent and -independent mechanisms [44]; this is accompanied by a reduction in β-adrenergic-mediated responses in hypertension [45], and a lower number of functional β2-adrenoceptors in PE [46]. Hence, the protective, anti-apoptotic effect of CST may be diminished in PE patients, resulting in a higher percentage of apoptotic cells. This effect may be strengthened by the impaired responses of the β2 adrenergic receptor. However, to determine whether CgA and CST have a direct impact on the apoptotic index in trophoblastic cells, further studies are necessary.

## 4. Materials and Methods

### 4.1. Cell Culture

The immortalized HTR-8/SVneo (CRL-3271) and BeWO (CCL-98) trophoblast cell lines were obtained from the American Type Culture Collection (ATCC) and maintained at 37 °C and 5% CO_2_, in Ham’s F-12K Nutrient Mix medium (Gibco Invitrogen, Carlsbad, CA, USA) and RPMI 1640 medium (Gibco Invitrogen, Carlsbad, CA, USA) supplemented with 2.5% Fetal Bovine Serum (FBS) (Gibco-BRL, Waltham, MA, USA) and 1% penicillin/streptomycin (Invitrogen, Waltham, MA, USA). For the experiment, the cells were seeded in a cell culture medium for 24 h in oxygen conditions typical of preeclampsia (2%O_2_) or healthy pregnancy (8%O_2_). Following this, the cells were transferred to a stimulating medium consisting of a medium typical for cell culture (F-12K or RPMI1640) together with 2.5%FBS, antibiotics and stimulants, i.e., IL6 or H_2_O_2_ or both (Table 1). This stimulating medium was replaced with a fresh one every 24 h for 72 h. For the final 24 h, the stimulating medium was deprived of FBS to minimize the contamination of the medium by catestatin, a natural component of serum.

All experiments were performed in 5% CO_2_ and 37 °C, under hypoxia (2%O_2_). The reference sample was incubated under normoxic conditions for placental cells, i.e., 8%O_2_. The experiments were conducted in InVivo2 Physio-logical Cell Culture Workstations (Baker Ruskinn, Sanford, ME, USA) to maintain constant oxygen and temperature conditions. All experiments were repeated three times (*n* = 3 experiments).

### 4.2. RNA Isolation

After 96 h of stimulation, the cells were washed with 1 mL of PBS solution (Gibco PBS, pH 7.2) and then treated with 800 µL of fenozol. Following this, total RNA was isolated with a Total RNA Mini Kit (A&A Biotechnology, Gdansk, Poland). The RNA quality and concentrations were measured on a NanoDrop spectrophotometer (Thermo Fisher Scientific, Warsaw, Poland).

### 4.3. CHGA Real Time PCR (RT-PCR)

cDNA was synthesized from 2000 ng of total RNA using a Maxima First Strand cDNA kit (Thermo Fisher Scientific, Warsaw, Poland). A RT-PCR reaction was performed with LightCycler 480 (Roche, Mannheim, Germany). Primers used for the reaction were commercially available and purchased from Qiagen (RT^2^ qPCR Primer Assay (200); Qiagen; RefSeq Accession no.: NM_001275.3). The analysis was performed with 1.5 µL of 10× diluted cDNA with RT^2^ SYBR Green Mastermix. The reactions were normalized with reference to the YWHAZ and GAPDH housekeeping genes, and a reference sample made from pooled samples (separate for each cell line). Normalized gene expression was calculated according to the Pfaffl equation [47].

### 4.4. Total Protein Extraction

The 0.25 × 10^6^ HTR-8/SVneo cell suspensions and 0.3 × 10^6^ BeWo cell suspensions were seeded in six-well plates in medium with FBS and incubated under hypoxic or normoxic (only controls) conditions for 24 h. Next, the stimulating medium was added (stimulation was performed as was described previously; Table 1, Section 4.1). After 96 h incubation, the cells were washed twice in dPBS (Gibco Invitrogen, Carlsbad, CA, USA) and treated with RIPA buffer (Sigma-Aldrich; Merck Life Science Sp.z.o.o., Poznan, Poland) with Halt™ Protease and Phosphatase Inhibitor Cocktail (Thermo Scientific, Waltham, MA, USA). The cells were transferred into chilled Eppendorfs and incubated on ice for 30 min. They were then centrifuged at 14,000 rpm for 10 min, at 4 °C. The supernatant (total protein fraction) was collected and stored at −80 °C for further analysis. Total protein concentration was measured using the BCA method (Pierce™ BCA Protein Assay Kit; Thermo Fisher Scientific, Warsaw, Poland).

### 4.5. Media Collection for Protein Analysis

The 0.25 × 10^6^ HTR-8/SVneo cell suspensions and 0.30 × 10^6^ BeWo cell suspensions were seeded in six-well plates in a medium with FBS and incubated under hypoxic conditions for 24 h. Next, the cells were stimulated by the addition of stimulating medium (as described previously; Section 4.1). After 72 h of stimulation, the cells were cultured for 24 h in 3 mL of the respective media without the addition of FBS, to avoid contamination with the CST naturally occurring in the serum.

### 4.6. Western Blot Analysis

The presence of CgA was confirmed using Western blot analysis. Following SDS-PAGE electrophoresis and PVDF membrane transfer, the membranes were blocked with 5% fat-free milk in TBST buffer (one hour, room temp). Next, the membranes were incubated overnight at 4 °C with CgA first primary antibody (Anti-CHGA antibody produced in rabbit, IgG fraction; SIGMA; Merck Life Science Sp.z.o.o., Poznan, Poland) in a 1:1000 dilution (antibody: 1% fat-free milk). GAPDH (MA5-15738, Thermo Fisher Scientific, Warsaw, Poland; produced in mouse) antibodies were used as an internal control at a 1:5000 dilution. The next day, the membranes were washed in TBST and incubated with APS conjugated secondary antibody (one hour, room temp), anti-rabbit (1:5000) and anti-mouse (1:10,000), respectively. The optical density of the visualized bands was analyzed using ImageLab software (Bio-Rad Laboratories, Hercules, CA, USA). Each experiment was replicated three times.

### 4.7. ELISA Test

The level of CST protein in the collected cell cultured media (Section 4.5.) was determined using Enzyme Linked-Immunosorbent Assay (ELISA). All procedures related to ELISA were performed according to the manufacturer’s instructions (RayBio^®^ Human/Mouse/Rat Catestatin Enzyme Immunoassay Kit; RayBiotech, Zory, Poland). Briefly, the plate was covered with the detection antibody and left overnight at 4 °C. Next, the samples were added. After incubation at room temperature (RT), streptavidin solution was added onto the plate, which was then incubated again at RT. Finally, the plate was covered with substrate reagent and left for 30 min. Following this, stopping solution was added and results were read at 450 nm.

The dilution factors for each analyzed protein were estimated experimentally, with a final decision of using non-diluted samples. The ELISA results were recalculated as nanograms of studied protein per 1 mL of cell culture media.

### 4.8. Apoptosis

The cells were seeded and cultured as described in Section 4.5. The Muse™ Annexin V with Dead Cell Assay Kit was used for the determination and differentiation of the four following distinct subpopulations of the cells localized in the one sample, i.e., live cells, early and late apoptotic as well as death cells. Briefly, following trypsinization and centrifugation, 1 × 10^5^ cells were placed in 100 μL of cell culture medium including FBS, and mixed with 100 μL of Muse™ Annexin V with Dead Cell Reagent (Merck Life Science Sp.z.o.o., Millipore, Poznan, Poland). This reagent includes fluorescently labeled Annexin V which connects to the phosphatidylserine being expressed outside of the cell membrane of apoptotic cells. The second fluorescent component of the Annexin V with Dead Cell Reagent, i.e., 7-aminoactinomycin D (7-AAD) labels the death and late apoptotic cells. After 20 min of incubation in the dark at room temperature, the measurement of apoptosis index was determined by the device recognized as miniaturized fluorescence detector and microcapillary cytometer, i.e., the Muse™ Cell Analyzer (Millipore, Germany).

### 4.9. Statistical Analysis

The obtained data were analyzed with the two-tailed Student’s *t*-test or ANOVA with the post hoc Bonferroni correction. The correlation between the catestatin level and apoptosis index were determined by the Spearman R test. All results were recognized as significant for a *p*-value < 0.05. The obtained results were analyzed by Statistica 13.1 (StatSoft, Krakow, Poland).

## 5. Conclusions

Our study provides the first evidence that CgA and CST proteins are produced by trophoblastic cell lines and that PE environment has an impact on CST protein production. Furthermore, a key finding was that a strong negative correlation appears to exist between CST protein level and apoptosis induction in trophoblastic cell lines.

Hence, it can be assumed that during PE, the placenta produces, and releases, higher amounts of CgA as a compensatory mechanism. However, this does not translate into higher levels of the protective peptide CST; in fact, the CST protein level is diminished, and the apoptosis of placental cells is increased. Our findings, and those of previous studies, suggest that both CgA and its derived peptide CST may play roles in the complex process of PE pathogenesis. However, subsequent experiments are necessary to confirm this.

## Figures and Tables

**Figure 3 ijms-24-07124-f003:**
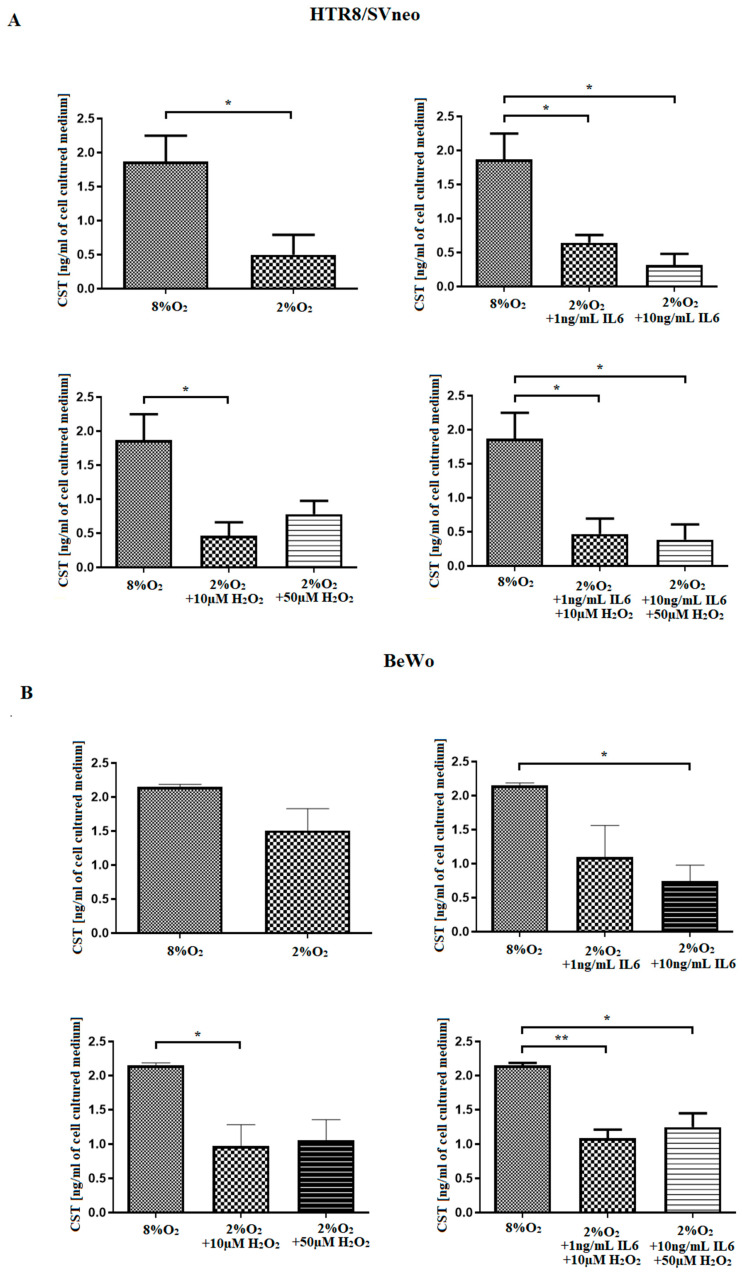
A comparison of CST secreted into the cell medium by hypoxic (2%O_2_) and normoxic (8%O_2_) trophoblastic cells, i.e., HTR-8/SVneo (**A**) or BeWo (**B**). CST level was determined in the cell culture medium to test whether CST is secreted by the cells into environment. Thus, the level of CST was determined as ng of studied protein included in 1 mL of cell cultured medium. The results are expressed as ± SEM; *n* = 3 experiments; the 8%O_2_ and 2%O_2_ cells were compared using Student’s *t*-test. The variants and controls were compared using one-way ANOVA with Bonferroni’s multiple comparison test. * *p* < 0.05 was considered as significant, ** *p* < 0.01 as compared to control.

**Figure 4 ijms-24-07124-f004:**
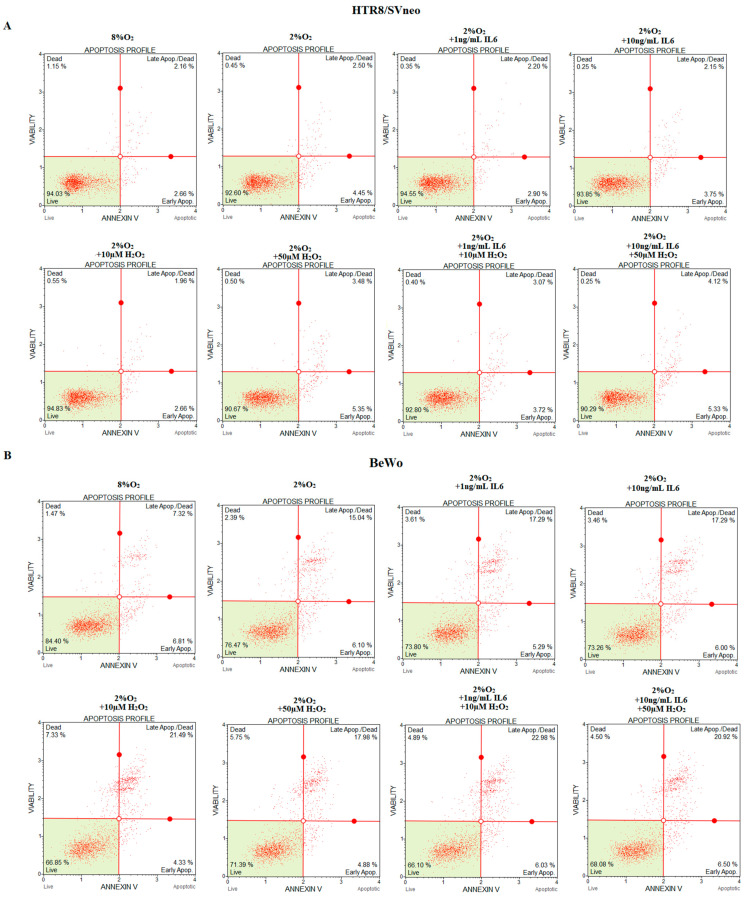
The representative results of the apoptosis profile obtained by flow cytometry with the Apoptosis Assay for Muse Cell Analyzer of differently stimulated HTR-8/SVneo (**A**) and BeWo (**B**) cells.

**Figure 5 ijms-24-07124-f005:**
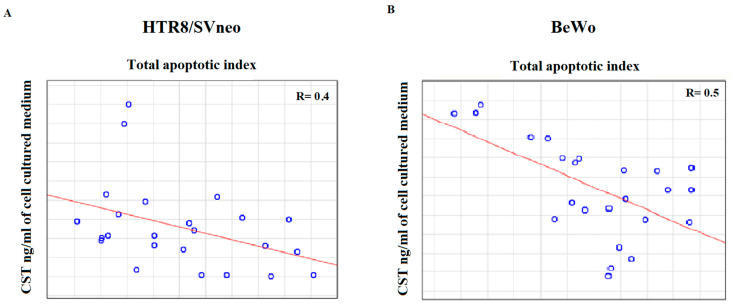
Spearman’s rank-order correlation between CST level and total apoptotic index (index for early and late apoptosis): (**A**) HTR-8/SVneo; (**B**) BeWo cell line.

**Table 1 ijms-24-07124-t001:** The stimulants and variants of the stimulation used in the study. The second variant refers to the different concentrations of the used factors.

Variants	Stimulants
Control	8%O_2_
Hypoxic conditions (HC)	2%O_2_
Inflammatory-like preeclamptic environment (ILPE1)	2%O_2_ +1 ng/mL IL6
Inflammatory-like preeclamptic environment (ILPE2)	2%O_2_ +10 ng/mL IL6
Oxidative stress-like preeclamptic environment (OLPE1)	2%O_2_ +10 µM H_2_O_2_
Oxidative stress-like preeclamptic environment (OLPE2)	2%O_2_ +50 µM H_2_O_2_
Preeclamptic-like environment (PLE1)	2%O_2_ +1 ng/mL IL6 10 µM + H_2_O_2_
Preeclamptic-like environment (PLE2)	2%O_2_ +10 ng/mL IL6 +50 µM H_2_O_2_

## Data Availability

Not applicable.

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
