# Peer review of "Reduction in CgA-Derived CST Protein Level in HTR-8/SVneo and BeWo Trophoblastic Cell Lines Caused by the Preeclamptic Environment"

_ijms, 2023, doi:10.3390/ijms24087124_

Round 1

Reviewer 1 Report

In this article, the role of chromogranin A (CgA) and its derived peptide, cates-tatin (CST), in the pathogenesis of preeclampsia (PE) is investigated. The study provides evidence that CgA and CST are produced by trophoblastic cell lines and reveals a negative correlation between CST protein levels and apoptosis induction.

However, there are some issues with the experimental design and display of the results. 

The author states on line 118 that each sample was tested three times, but the provided Western blot image seems to only show two duplicates. 

To avoid non-specific protein changes during apoptosis, it is important to separate apoptotic cells from normal cells during analysis. 

The article also lacks information on the species (mouse or rabbit) of the "GAPDH antibodies" on line 388.

The original Western blot image was stained with two antibodies simultaneously, but the article incorrectly states "or" instead of "and." (Line 291)

There are missing information in reference 4, 6, 8 and 29.

Figure 4 does not clearly indicate the concentration of the ELISA experimental results.

While the study can simulate the environment of oxidative stress and pre-eclampsia, it is possible that the results are directly related to cell death. Protein degradation during apoptosis is non-specific, and therefore, the current conclusion that CgA and CST are related to hormonal changes secreted by trophoblast cells may be premature.

Author Response

Reviewer 1:

Comments and Suggestions for Authors

  1. The author states on line 118 that each sample was tested three times, but the provided Western blot image seems to only show two duplicates. 

 Authors’ answer:

Presented Western blot image was named as ‘representative’ as it depicts only one replication of each sample (e.g. PLE1 and PLE2 refers to different stimulation variants as described in Table 1.). However, in the end each variant was tested three times, for each cell line. To more determine the types of stimulations of HTR8/SVneo cells presented at the Western blot, we add to the Legend the description of each acronym presented on the picture. Further, additional description has been added into the Results section and respective acronyms were added below the graphs in Fig.1.

  1. To avoid non-specific protein changes during apoptosis, it is important to separate apoptotic cells from normal cells during analysis. 

 Authors’ answer:

To determine the profile of apoptosis, we used the Muse® Annexin V & Dead Cell Assay Kit (https://www.merckmillipore.com/PL/pl/20151006_191750 ). This kit allows determining the percent of live, early apoptotic, late apoptotic as well as death cells. Therefore the separation of apoptotic cells from live (i.e. normal) cells during analysis is prohibited to the impossibility of determination of each subpopulation of the cells analysed by the Annexin V & Dead Cell Assay Kit. These fractions are visible of the flow cytometry graphs presented as a Fig.5.

To presents the background of the Muse® Annexin V & Dead Cell Assay Kit we modified the 4.8. Subsection of Apoptosis in the Material and Methods chapter:

4.8. Apoptosis

The cells were seeded and cultured as described in section 2.5. The Muse™ Annexin V with Dead Cell Assay Kit was used for the determination and differentiation of the four following distinctsubpopulationsof the cells localized in the one sample i.e. live cells, early and late apoptotic as well as death cells. Briefly, following trypsinization and centrifugation, 1x10^5cells were placed in 100ul of cell culture medium including FBS, and mixed with 100ul of Muse™ Annexin V with Dead Cell Reagent (Millipore, Germany). This reagent includes fluorescently labeled Annexin V which connect to the phosphatidylserine being expressed outside of the cell membrane of apoptotic cells. The second fluorescent component of the Annexin V with Dead Cell Reagenti.e. 7-aminoactinomycin D (7-AAD) labels the death and late apoptotic cells. After 20 minutes of incubation in the dark at room temperature, the measurement of apoptosis index was determined by the device recognized as miniaturized fluorescence detector and microcapillary cytometer i.e. theMuse™ Cell Analyzer (Millipore, Germany).

  1. The article also lacks information on the species (mouse or rabbit) of the "GAPDH antibodies" on line 388.

 Authors’ answer:

The proper species has been added as follows:‘…GAPDH (MA5-15738, Thermo, USA;produced in mouse) antibodies…’

  1. The original Western blot image was stained with two antibodies simultaneously, but the article incorrectly states "or" instead of "and." (Line 291).

 Authors’ answer:

The authors used ‘or’ to stress the difference between two different secondary antibodies used. Yet, if such notations misled the reader, the authors agree with a suggestion to change it:

‘…anti-rabbit (1:5000) and anti-mouse (1:10 000) respectively…’

  1. There are missing information in reference 4, 6, 8 and 29.

 Authors’ answer:

The Authors apologize for this mistake. Proper changes have been made.

  1. Figure 4 does not clearly indicate the concentration of the ELISA experimental results.

Authors’ answer:

We corrected the description of Y axis (CST [ng/ml of cell cultured medium]. CST level was tested in the cell culture medium to determine whether CST is secreted by the cells into environment. Thus, the level of CST was determined as ng of studied protein included in 1 ml of cell cultured medium.

  1. While the study can simulate the environment of oxidative stress and pre-eclampsia, it is possible that the results are directly related to cell death. Protein degradation during apoptosis is non-specific, and therefore, the current conclusion that CgA and CST are related to hormonal changes secreted by trophoblast cells may be premature.

Authors’ answer:

We agree with the Reviewer that it is not clear whether the changes in CST level are related to the preeclamptic environment or it is result of degradation of proteins e.g. CgA in the apoptotic process. However, we observed the significant changes in secretion of CST level between HTR8/SVneo cells incubated in normoxia and hypoxia with 1ng/ml IL6 or 10ng/ml IL6 or 10 µM H2O2, but for these variants of stimulation the differences in apoptotic processes were not observed. Therefore, we believe that the lower secretion of CST into environment was not the result of apoptotic process but a natural reaction of trophoblastic cells on the various variants of stimulation mimicking the preeclamptic conditions. We add the following information into discussion section:

However, to determine whether CgA and CST have a direct impact on the apoptotic index in trophoblastic cells, the further studies are necessary.

Further, because the direct association between apoptosis and environment is not so important for the results of our study as important is the correlation, therefore we removed the graphs that presented the apoptotic index of the cells in preeclamptic conditions as the results were described in the text. We keep in the Fig 5 only the raw data of cytometric analyses.

Reviewer 2 Report

Preeclampsia (PE) is the most severe complication in the pregnancy that is associated with high blood pressure and renal dysfunction, and may result in fetal growth restriction. The etiology of PE has not been fully understood. In the current manuscript, the authors focused on chromogranin A (CgA) and its derivative catestatin (CST). CgA and CST are produced by chromaffin cells of the adrenal medulla and have been implicated in the occurrence of high blood pressure. They are also thought to be an anti-apoptotic factor in several cancers. Although the placenta expresses CgA, whether trophoblasts in the placenta express CgA and the effects of pathological conditions on CgA expression have been unclear. In the current study, they provide evidence that trophoblast cell lines expressed CgA and the CgA expression may be reduced in “preeclamptic conditions”. The manuscript needs to be revised before acceptance.

1. Regarding the hypoxic condition in the HTR8 experiments, the first trimester is considered to be hypoxic, which may promote invasion of extravillous trophoblasts and formation of spiral arteries. In the manuscript, the authors showed that culturing HTR8 cells in a hypoxic condition reduced CgA expression, which seemed to be pathological and need an explanation.

2. Fig. 2 and Fig. 3, abbreviations in Fig.3 do not exactly correspond to those of Table 1. The authors used some “preeclamptic conditions” such as hypoxic, inflammatory, and oxidative conditions. Please cite appropriate references showing that these conditions are actually “preeclamptic”. If the graphs of Fig. 2 are calculated based on the immunoblot of Fig. 3, these figures may be combined as a single figure. In Fig. 3, the control CgA band intensity seems to be lower than that of ILPE1 and OLPE1, which raises a doubt against the quantification shown in Fig. 2. Molecular weights must be indicated in Fig. 3.

3. The link between CgA/CST levels and induction of apoptosis is not clearly shown by the data. They show a correlation between them in Fig. 6 (not referred in the main text), but the possibility that “preeclamptic conditions” induced apoptosis independently on reducing CgA/CST levels cannot be ruled out. If the authors want to state that actually there is a correlation between CgA/CST levels and apoptosis induction by “preeclamptic conditions”, the apoptosis under the “preeclamptic conditions” must be rescued by exogenous CgA and/or CST. Enhanced apoptosis in PE placentas should be shown or at least appropriate refs must be cited. The description of CgA and apoptosis, at least in the conclusion, should be toned down.

4. Discussion is redundant, please focus on the possible pathological and physiological roles of CgA and CST in the placenta.

Minor

1. Il6 should be IL-6 or IL6.

2. The authors used the term “depleted”, but the data show “reduced”.

3. Please check the refs throughout the manuscript. For example, Ref 11 does not seem to be published in 2094. Ref 44 on line 313 should be Ref 43 on the ref list.

4. Line 368, please refer Table 1 for “stimulation was performed …”.

5. Line 388, please provide information of the GAPDH antibody.

6. Please check typos throughout the manuscript.

Author Response

Reviewer 2

Comments and Suggestions for Authors

  1. Regarding the hypoxic condition in the HTR8 experiments, the first trimester is considered to be hypoxic, which may promote invasion of extravillous trophoblasts and formation of spiral arteries. In the manuscript, the authors showed that culturing HTR8 cells in a hypoxic condition reduced CgA expression, which seemed to be pathological and need an explanation.

Authors’ answer:

We want to thank the Reviewer for this question. According to the obtained outcomes, CgA expression was significantly reduced only in two stimulating variants (mixes of IL6 and H2O2 in a different concentrations), while hypoxia itself resulted in the reduction of CgA protein level (both only in HTR8/SVneo cells).

It is true that it is very difficult to determine the role of CHGA at the beginning of gestation. It is possible that the low oxygen level and inflammatory condition typical for the first stages both preeclamptic and non-complicated gestation negatively regulates the expression of chromogranin A. It might support the process of shaping ofimmunological balance in decidua necessary for the correct course of gestation; indeed depletion in chromogranin A regulates the macrophage function, and reduce their chemotaxis activity (Eissa, N. , Hussein, H. , Kermarrec, L. , Ali, A. Y. , Marshall, A., Metz‐Boutigue, M. , Hendy, G. N. , Bernstein, C. N. , & Ghia, J. E. (2018). Chromogranin‐A also regulates macrophage function and its apoptotic pathway, as was observed in murine DSS colitis. Journal of Molecular Medicine, 96, 183–198. 10.1007/s00109-017-1613-6) However, with the course of gestation when the environment is changing from hypoxia into normoxia the level of chromogranin A should elevate to support the process of placental cell proliferation and to deliver the chromogranin A derivative peptides (i.e. CST) supporting the maternal blood pressure regulation. It is also possible, that in case of PE pregnancies level of CHGA expression may raise even more, comparing to the healthy pregnancies, which could be a part of compensatory mechanism of the maternal organism. However, the level of CgA protein does not differ at the late gestation between preeclamptic and normotensive placental cells, as was presented in the present study (results for BeWo cell line) and as was determined for placental samples (results published by our research team: Bralewska M, Biesiada L, Grzesiak M, Rybak-Krzyszkowska M, Huras H, Gach A, Pietrucha T, Sakowicz A (2021) Chromogranin A demonstrates higher expression in preeclamptic placentas than in normal pregnancy. BMC Pregnancy Childbirth 2021;21:1–10. doi: 10.1186/s12884-021-04139-z.). This suggests that the level of CgA might be regulated by the posttranslational processes or in preeclampsia the high level of full peptide of CgA is immediately digested into not determined in this study CgA-derived peptides, thus the level of full peptide of CgA does not differ between preeclamptic and noncomplicated gestation. Considering above, the following explanation for the obtained results was given in the discussion section:

‘As mentioned before, at the beginning of pregnancy (here represented by the HTR-8/SVneo cells), low oxygen level and mild inflammation are necessary for proper implantation. Therefore, the fact that significantly lower CgA protein level was observed between cells incubated in 8%O2 and 2%O2 suggests that trophoblastic cells both from early normotensive and preeclamptic pregnancy characterize low production of CgA. This low CgA production by early trophoblast is additionally depleted in hypoxia complicated by inflammatory conditions (ILPE1 i.e. 2%O2+1ng/ml IL-6+10µM H2O2 and ILPE2 i.e. 2%O2+10ng/ml IL-6+50µM H2O2) in comparison to 8%O2. However, this level is insignificant lower between cells incubated only in 2%O2 vs ILPE1 or 2%O2 vs ILPE2 (do not shown results).
Furthermore, in the present study, CHGA expression and CgA protein level did not significantly differbetween the variants representing the third trimestertrophoblast i.e. BeWo cell line. Interestingly, our previous study conducted on placental samples obtained at the term of gestation found only CHGA expression level, but not protein, elevated in PE patients compared to healthy controls [29]. This may suggest that together the course of preeclamptic pregnancy, maternal organism attempts to increase the level of CgA protein by overexpression of its gene, however this mechanism might be ineffective or the full protein CgA is immediately digested into smaller peptides that can lead to the adverse pregnancy outcomes.‘

  1. A) Fig. 2 and Fig. 3, abbreviations in Fig.3 do not exactly correspond to those of Table 1. B) The authors used some “preeclamptic conditions” such as hypoxic, inflammatory, and oxidative conditions. Please cite appropriate references showing that these conditions are actually “preeclamptic”.
    C) If the graphs of Fig. 2 are calculated based on the immunoblot of Fig. 3, these figures may be combined as a single figure. In Fig. 3, the control CgA band intensity seems to be lower than that of ILPE1 and OLPE1, which raises a doubt against the quantification shown in Fig. 2. Molecular weights must be indicated in Fig. 3.

Authors’ answer:

  1. The authors apologize for this error, the abbreviation typos have been corrected where necessary. Further, full description of the acronyms was added into the legend.

  1. Following explanations were given in the Introduction and Results sections:

,One of the most widely- accepted theoriesof PE proposes that shallow trophoblast invasion into maternal decidua forces placental cells to grow in hypoxic conditions (1-2% of oxygen), leading to uncontrolled placental apoptosisand fetal malnutrition[2].This results in the appearance of multiple placental factors, such as soluble fms-like tyrosine kinase 1 (sFlt-1) or soluble endoglin (sEng), in the maternal circulation[3].These factors act on the maternal vascular endothelium, inducing oxidative stress and the production of reactive oxygen species (ROS), e.g. superoxide (O2•−) and hydrogen peroxide (H2O2); they also stimulate the production and secretion of pro-inflammatory cytokines (e.g. tumor necrosis factor α; TNFα, interleukin 6; IL-6) into the maternal blood [2, 4, 5].’

and

Preeclamptic conditions were replicated using two stimulants, viz.Il-6 and H2O2, reflecting the respective inflammation and oxidative stress present in the PE placenta. The stimulants were chosen based on previous studies indicating that both are known to be elevated in PE patients [26, 27]. Hypoxia was generated in a hypoxia chamber, ensuring constant, long-lasting culture in 2% of oxygen. Precise selection of the oxygen conditions is of key importance to the whole study. As noted by PavlackyandPolak, an improper oxygen level can result in various multiple cellular responses influencing cell signaling, migration, proliferation and apoptosis[28]. Therefore, instead of the 21% oxygen concentration typically usedfor this experiment, 8% O2 was used for the normoxic controls, as this value corresponds to the physiological trophoblast oxygen level [17]

  1. C) Fig 2 and 3 were combined as suggested. Regarding intensity of the control band: the presented outcome is only representative as it depicts just one repetition. Therefore presentation of the molecular weight of this one repetition (out of three) could be misleading. Further, the final relative CgA protein level was calculated by comparison to the GAPDH (reference protein) level which was visualized separately for each variant. Summing up, the final relative CgA protein level, e.g. for the control sample, was obtained after average of three different outcomes, which were previously compared to their respective references (each sample had its own GAPDH reference). Therefore, it is possible that the presented visualization may slightly vary from the numerical data; however it is a part of the entire experiment and the authors would like to keep it in the presented form. The authors hope that the Reviewer will positively consider this request.

The representative data used for relative CgA protein level calculation are presented below (calculations are for the representative gel presented in the article):

Band No.

Band Label

Mol. Wt. (KDa)

Adj. Volume (Int)

Volume (Int)

Abs. Quant.

ratio CgA/GAPDH

sample

1

CgA

N/A

0,202703

59695088

263161076

N/A

0,380802067

HC

1

GAPDH

N/A

0,511261

156761460

576024708

N/A

2

CgA

N/A

0,238739

25088115

109612620

N/A

0,212822939

PLE2

2

GAPDH

N/A

0,524775

117882570

434888610

N/A

3

CgA

N/A

0,247748

30975750

155118834

N/A

0,265556761

PLE1

3

GAPDH

N/A

0,533784

116644554

413200710

N/A

4

CgA

N/A

0,243243

39930085

203899978

N/A

0,39778211

OLPE2

4

GAPDH

N/A

0,547297

100381802

511501774

N/A

5

CgA

N/A

0,297297

87656030

266745710

N/A

0,497964643

OLPE1

5

GAPDH

N/A

0,578829

176028622

477922654

N/A

6

CgA

N/A

0,295045

104234689

385958768

N/A

0,533310998

ILPE1

6

GAPDH

N/A

0,581081

195448227

648955281

N/A

8

CgA

N/A

0,272523

86701659

259688834

N/A

0,49809531

ILPE2

8

GAPDH

N/A

0,567568

174066403

448459853

N/A

9

CgA

N/A

0,272523

94720002

326936704

N/A

0,602214554

control

9

GAPDH

N/A

0,565315

157286139

400830485

N/A

  1. The link between CgA/CST levels and induction of apoptosis is not clearly shown by the data. They show a correlation between them in Fig. 6 (not referred in the main text), but the possibility that “preeclamptic conditions” induced apoptosis independently on reducing CgA/CST levels cannot be ruled out. If the authors want to state that actually there is a correlation between CgA/CST levels and apoptosis induction by “preeclamptic conditions”, the apoptosis under the “preeclamptic conditions” must be rescued by exogenous CgA and/or CST. Enhanced apoptosis in PE placentas should be shown or at least appropriate refs must be cited. The description of CgA and apoptosis, at least in the conclusion, should be toned down.

Authors’ answer:

We agree with the Reviewer, that the influence of low level of CHGA or CST on the apoptotic index of trophoblastic cells was not determined in this study. However, the present study does not indicate the association but the correlation. Therefore, we present only our observations that the correlation between the low level of CST and the high apoptotic index exists. It is true that other factors might influence our observations, i.e. elevated degradation of protein including CST might be a reason of apoptotic process, as was suggested by the Reviewer 1. However, we observed the significant changes in secretion of CST level between HTR8/SVneo cells incubated in normoxia and hypoxia with 1ng/ml IL6 or 10ng/ml IL6 or 10 uM H2O2, but for these variants of stimulation the differences in apoptotic processes were not observed. Therefore, we believe that the higher secretion of CST into environment was not the result of apoptotic process but a natural reaction of trophoblastic cells on the various variants of stimulation mimicking the preeclamptic conditions. We add the following information into discussion section:

However, to determine whether CgA and CST have a direct impact on the apoptotic index in trophoblastic cells, the further studies are necessary.’

Further, because the direct association between apoptosis and environment is not so important for the results of our study as important is the correlation, therefore we removed the graphs that presented the apoptotic index of the cells in preeclamptic conditions as the results were described in the text. We keep in the Fig 5 only the raw data of cytometric analyses.

  1. Discussion is redundant, please focus on the possible pathological and physiological roles of CgA and CST in the placenta.

Authors’ answer:

The authors have removed from the discussion section a fragment describing the preeclampsia pathogenesis. Further, following fragment was transferred from the Discussion to the Results section:

‘Preeclamptic conditions were replicated using two stimulants, viz.Il-6 and H2O2, reflecting the respective inflammation and oxidative stresspresent in the PE placenta. The stimulants were chosen based on previous studies indicating that both are known to be elevated in PE patients [26, 27]. Hypoxia was generated in a hypoxia chamber, ensuring constant, long-lasting culture in 2% of oxygen. Precise selection of the oxygen conditions is of key importance to the whole study. As noted by PavlackyandPolak, an improper oxygen level can result in various multiple cellular responses influencing cell signaling, migration, proliferation and apoptosis [28]. Therefore, instead of the 21% oxygen concentration typically usedfor this experiment, 8% O2 was used for the normoxic controls, as this value corresponds to the physiological trophoblast oxygen level [17]. Additionally, instead of short-term stimulation, the cultures were maintained for 96 hours to best imitate the process ongoing in the PE placenta. Each 24 hours of culture in stimulating medium corresponds to ten weeks of gestation; therefore, the 96hours used in this study correspondstothe natural 40-weekduration of gestation.’

Minor

1. Il6 should be IL-6 or IL6.

- proper changes have been made; the acronym IL6 has been used

  1. The authors used the term “depleted”, but the data show “reduced”

- the term ‘depleted’ was replaced by ‘reduced’

  1. Please check the refs throughout the manuscript. For example, Ref 11 does not seem to be published in 2094. Ref 44 on line 313 should be Ref 43 on the ref list.

- proper changes have been made

  1. Line 368, please refer Table 1 for “stimulation was performed …”

- reference was added

  1. Line 388, please provide information of the GAPDH antibody

- following information has been added: GAPDH (MA5-15738, Thermo, USA;produced in mouse)

  1. Please check typos throughout the manuscript.

- proper correction have been made where necessary

Round 2

Reviewer 1 Report

All mistakes and unacceptable presentations have been fixed.

Reviewer 2 Report

The revised manuscript has been sufficiently improved.